# Equimolar Polyampholyte Hydrogel Synthesis Strategies with Adaptable Properties

**DOI:** 10.3390/polym15143131

**Published:** 2023-07-23

**Authors:** Gaukhar Toleutay, Esra Su, Gaukhargul Yelemessova

**Affiliations:** 1Laboratory of Engineering, Satbayev University, Almaty 050013, Kazakhstan; 2Department of Chemistry, University of Tennessee, Knoxville, TN 37996, USA; 3Aquatic Biotechnology, Faculty of Aquatic Sciences, Istanbul University, 34134 Istanbul, Turkey

**Keywords:** polyampholytes, electrostatic interactions, H-bonds, mechanical properties

## Abstract

Polyampholyte hydrogels exhibit great antibacterial and antifouling properties, which make them attractive for biomedical applications, such as drug delivery, wound healing, and tissue engineering. They also have potential applications in food safety, wastewater treatment, and desalination. Since they are based on ionic interactions, polyampholytes are known to require lower amounts of chemical cross-linkers as compared with traditional gels. However, the effects of both chemical and physical interactions on the material’s performance are yet to be fully understood and were examined in the present work. Here, four series of equimolar polyampholyte hydrogels were synthesized with anionic (acrylamidomethylpropane sulfonic acid sodium salt) and cationic monomers (acrylamidopropyl-trimethylammonium chloride) along with a cross-linker (N,N′-methylenebisacrylamide). The mechanical and rheological properties of the gels were characterized following changes to the initial monomer concentration and crosslinker ratios, which led to gels with different toughness, stretchability, and compressibility. The direct correlation of the cross-linking degree with the initial monomer concentration showed that the chemical crosslinker could be further reduced at a high monomer concentration of 30% by weight, which creates an inter-chain network at a minimal crosslinker concentration of 0.25%. Lastly, N′N-dimethylacrylamide was added, which resulted in an increase in the number of H-bonds in the structure, noticeably raising material performance.

## 1. Introduction

Polyampholyte (PA) hydrogels belong to the class of polyelectrolytes containing negatively and positively charged functional units in the macromolecular chain and are divided into the following subgroups: weakly charged, strongly charged, and polymeric betaines, also called zwitterionic polyampholytes [1,2,3]. Hydrogels made from charged units have many unique properties, such as their ability to respond to changes in pH and salinity, similarity to certain biological systems, low toxicity, and biocompatibility. As such, they have many applications, including antibacterial, antifouling, and salt-resistant materials, among others. They can also be combined with other compounds such as polyacrylamide and polyethylene glycol to further enhance their mechanical properties, including toughness, elasticity, and adhesion [4,5,6,7,8,9,10].

In recent years, the main attention has been focused on equimolar PA and polyionic complex (PIC) hydrogels [11,12]. A series of experiments were carried out by Gong et al. [11,13,14,15,16,17,18]. They fabricated very rigid PA copolymeric hydrogels based on (methacryloylaminopropyl)trimethylammonium chloride (MAPTAC) and 4-vinylbenzenesulfonate (NaSS) (P(MAPTAC-co-NaSS)) using high concentrations of oppositely charged ionic monomers via free radical copolymerization. The structural and morphological differences between these systems lead to variations in mechanical properties. Composed of both permanent and reversible electrostatic interactions and hydrogen bonds, the mechanical properties of these hydrogels can be reversibly regulated by pH and ionic charge. Also, these bonds arise through both inter-chain and intra-chain interaction, helping to distribute the strain under stress more evenly throughout the material, resulting in increased durability [13,14,15,16].

Again, in a study presented by the Gong group, polyionic complex (PIC) hydrogels exhibit extremums near the charge balance point, where Coulombic attraction prevails over repulsion, resulting in collapsed polymer chains. The mechanical strength increases dramatically at the charge balance point [17]. A detailed review by Kudaibergenov and Okay showed that the PIC hydrogels are much tougher than the equimolar PA hydrogels. Also, the PIC hydrogel shows a more inhomogeneous structure with a broad pore size distribution (0.5–3.0 μm), while the equimolar PA hydrogel shows a more homogeneous structure (pore sizes in diameter are 0.1–0.3 μm) [12].

In previous works from our group, we compared the effects of varying the monomer concentration of a polymapholyte hydrogel to the effects of mechanical and swelling properties. It was shown that swelling volume ratio, YounG’s modulus, and compressive fracture stress of hydrogels are strongly affected by charge fraction, which agrees with Gong’s and coauthors’ findings [19]. In another work, we reported the synthesis and characterization of strongly charged PA hydrogels with linear and cross-linked structures. Detailed investigations have been made on the linear and gel behavior of polymers, including the formation of complexes of PA hydrogels with ionic surfactants and dyes [20].

We also synthesized a new class of thermally healable, hydrophobically modified equimolar PA hydrogels. PAMPS-PAPTAC (polyacrylamido-2-methyl-1-propanesulfonic acid-polyacrylamidopropyl-trimethylammonium chloride) gels at various concentrations in the presence of MBAA methylenebis(acrylamide), a chemical crosslinker, together with a hydrophobic monomer, n-octadecyl acrylate (C18A), and studied the rheological and mechanical properties of the PA hydrogels [20,21]. More studies are needed to further understand the design principles of these materials and how to optimize them for specific applications.

Now, the properties of charge-balanced PAMPS-PAPTAC and PAMPS-PAPTAC/DMAA hydrogels are discussed. These were synthesized via thermal polymerization with a total monomer concentration fixed at 5 and 30 wt%, and the amount of chemical crosslinker varied between 20 and 0.1 mol%. The resultant PA hydrogels were examined to ascertain their swelling and mechanical characteristics, and the effects of the hydrophilic N, N-dimethylacrylamide (DMAA) monomer on mechanical and rheological performance were studied. Unlike other studies here, we investigate the effect of additional H-bonding groups as well as ionic interactions to replace polyampholyte hydrogels formed using hydrophobic and ionic interactions.

## 2. Materials and Methods

All chemicals, including acrylamidopropyl-trimethylammonium chloride (APTAC), 50 wt% in water; sodium salt acrylamidomethylpropane sulfonic acid (AMPS), 75 wt% in water; N,N-dimethylacrylamide (DMAA, 99%); N,N′-methylenebis(acrylamide) (MBAA); and ammonium persulphate (APS), were supplied by Sigma-Aldrich (St. Louis, MO, USA) and used without further purification.

### 2.1. Preparation of PA Hydrogels

Four sets of PA hydrogels were synthesized via thermal polymerization of the AMPS, APTAC, and DMAA monomers at 60 ± 2 °C using the chemical crosslinking agent N,N′-methylenebisacrylamide and 10 mM of initiator Ammonium persulfate. In the first 2 sets of synthesizing hydrogels, the equimolar PAMPS-PAPTAC composition hydrogels were synthesized with total monomer concentrations in a range of 5–30 wt%, and the chemical crosslinker also varied from 20 to 0.1 mol.%. In the 3rd set, the total monomer concentration was fixed at 30 wt% and the chemical crosslinker varied from 5 to 0.25 mol.%, respectively. In the 4th set, the hydrophilic monomer DMAA was added to the mixture and ranged from 5 to 20 mol.% (Table 1).

### 2.2. Swelling Tests

Gel fraction, *W_g_*, was obtained by the $W_{g}=\frac{m_{dry}}{C_{M}\times m_{0}}$ formula, where *m*_0_ and *m_dry_* are the after prepared and dried hydrogel weights, respectively, *C_M_*, wt% is the concentration of monomer. The swelling ratios, m_rel_, of after prepared gels were determined with the formula: $m_{rel}=\frac{m_{swl}}{m_{0}}$ where *m_swl_* is the weight of the swollen gel specimen.

### 2.3. Rheological Measurements

Rheological measurements were conducted with the Bohlin 150 rheometer system (Malvern Instruments, Malvern, UK) with a Peltier sensor capable of regulating the temperature between 0 and 80 °C. The hydrogel samples were placed among rheometer panels at a steady distance of 150 mm at an ambient temperature of 25 °C in order to monitor gel formation time, where frequency varied from 0.1 to 100 rad·s^−1^ and strain amplitude γ_0_ was equal to 0.01. The gel formation time was monitored for 1 h at 60 °C, after which the hydrogel samples were removed from the rheometer.

### 2.4. Mechanical Measurements

A 500 N load cell was used in the compression measurements, which occurred at ambient temperature on a ZwickRoell testing system (Kennesaw, GA, USA). To establish full contact between the sample and the surface, 0.05 N was applied as a pre-load. Cylindrical gel specimens were utilized in the tests, and a fixed crosshead rate of 3 mm/min^−1^ was applied for collecting load and displacement data. During deformation, sample volumes were assumed to remain constant. The deformation ratio was used to measure the actual area of the specimen between the deformed length and the original length. In the case of undeformed and deformed specimens, the nominal σ_nom_ and true σ_true_ values of compression stress can be referred to as force per crossover area. The nominal stress is the theoretical stress that is calculated based on the applied force. The true stress is the stress that is actually experienced by the specimen due to deformation and is usually higher than the nominal stress. The strain is expressed as the deformation ratio (or by the biaxial extension ratio λ_biax_ = λ^−0.5^). During cyclic tensile experiments, a maximum strain of 300%, depending on the sample, was observed, followed by immediate release to zero displacement, with measurements being taken after waiting periods of 1 and 2 min for each sample, resulting in five cycles.

## 3. Results and Discussion

In our previous works, free radical copolymerization of PAMPS and PAPTAC with MBAA as the crosslinking agent has been investigated to determine the mechanical properties of PA hydrogels. As a result of these studies, it was revealed that the mechanical properties of hydrogels are greatly improved in equimolar monomer composition compared to PA hydrogels containing more cationic or anionic monomers due to electrostatic interactions [18,19,20]. Hence, we have now synthesized three series of equimolar PA hydrogels with different monomers/crosslinking agent ratios, aiming at improving their flexibility Therefore, in Series A, hydrogels with an initial monomer concentration (C_M_) of 5 wt% were synthesized, while the amount of crosslinking agent MBAA varied from 20 to 5 mol.%. In Series B, the amount of MBAA was adjusted to 5 mol.% in moles, while the C_M_ varied between 5 and 30 wt%. In Series C, C_M_ was fixed at 30 wt%, and the MBAA content varied between 5 and 0.1 mol.%. In Series D, the monomer concentration C_M_ and chemical crosslinker concentration were kept constant at 30 wt% and 0.5 mol.%, and DMAA, which is the uncharged hydrophilic monomer, was added to the structure (Table 1). Hereafter, the samples will be referred to according to their C_M_ and the crosslinker concentration as Cx/y. For example, a sample with 5 wt% of monomers and 20 mol.% of MBAA will be referred to as C5/20. In Serie D, monomer concentration and crosslinker are fixed as C30/0.5, so DMAA was added as an additional index in the sample nomenclature; they are designated as C30/0.5/z (Table 1). The structure and content for the synthesis of PA hydrogels are presented in Figure 1 and Table 1.

### 3.1. Series A and B

First, the rheological behavior of polymers, including loss (G″) and storage (G′) moduli, was investigated (Figure 1 and Figure 2). In the figures, tan δ = G″/G′ represents the loss factor, that is, the strain lost due to internal friction. The gelation point, *t_gel_*, appears at the intersection of G′ and G″, representing the transition back to a solid state following viscous behavior, and is considered to be an indicator of the gelation time of a hydrogel. The sensitivity limit of the rheometer is thought to be associated with a strong change in G″ at the beginning of the measurement period. Following the induction time, G″ becomes insignificant, and, as a result, G′ increases rapidly.

The value of the gelation point, *t_gel_*, is based on the rate of polymerization and the amount of monomers, cross-linker, and initiator. Monitoring the polymerization reaction helps determine the rate of crosslinking as a function of the concentration of the crosslinker.

As shown in Figure 1a, the sample C5/5 presented no gelation, as characterized by the absence of changes in G′ or G″ and, therefore, the nearly constant behavior of tan δ. The gelation process for the C5/20 hydrogel, as shown in Figure 1c, is characterized by a sharp increase in G′ to 2.8 kPa, thus demonstrating that the hydrogel is stronger than the C5/10 variant (Figure 1b). Further, tgel for samples C5/20 and C5/10 is reached at an early stage of polymerization, being 9.6 and 31.1 min, respectively. Overall, these results show that C5/5 presents an unstable viscoelastic state during the entire polymerization time, which is explained by an underdeveloped polymer matrix as a result of elastically inefficient dangling ends and loops. Also, the gelation times of the hydrogels are directly dependent on the initial concentrations of the crosslinking agent and the monomers. That is, a higher content of the crosslinker results in a shorter gelation time, as summarized in Table 2.

The range of linear viscoelastic dependence is characterized by the independence of G′ from the strain amplitude. That range over which deformation of a sample can occur without breaking is equated to impact strength. A declining G′, as seen in the graphs, indicates brittle fracturing of the hydrogels. At the point of intersection, where G′ = G″, microcracks appear in hydrogel samples. However, if G′ > G″, microcracks grow into continuous macrocracks throughout the samples, thus demonstrating a loss in ductile behavior.

In a subsequent experiment, the samples from Series A and B were placed between the rheometer plates for 1 h at 60 °C, and the collected results are shown in Figure 2. In this case, G′ of the hydrogels C5/20 and C5/10 does not depend on frequency (Figure 2b,c), and tan δ remains below 0.1 (see inset), corresponding to the behavior of hydrogels in a solid state. The supramolecular C5/5 hydrogel (Figure 2a), in contrast, has a frequency-dependent storage modulus, and tan δ remains above 0.1 throughout the experiment.

The graphs in Figure 1 show that with increasing monomer concentration, there is also a tendency for gelation time *t_gel_* to decrease. In the case of sample C15/5, *t_gel_* reached 21 min, whereas for sample C30/5 it is only 78 s. Meanwhile, the loss factor increased to 3.0 for the C15/5 sample and to 10.4 for the C30/5 sample. The viscoelastic spectra of C15/5, C30/5, and C5/5 hydrogels are shown in Figure 2. The loss factors for C15/5 and C30/5 were below 0.1, specifically 6 × 10^−4^ for C30/5 and 1 × 10^−3^ for C30/5, which correspond to the solid behavior of chemically crosslinked PA hydrogels. Further, the storage moduli remain unchanged at the values of G′ = 9.0 and 25.5 kPa for C15-5 and C30-5, respectively. However, with an increase in monomer concentration, as in C15/5 and C30/5, the viscosity of the hydrogels tends to increase. Figure 1 and Figure 2 show that, with increasing monomer concentration, the G′ of the hydrogels also increases. For the C5/5 sample, the storage modulus is a mere 0.1 kPa, whereas for the C15/5 and C30/5 hydrogels, the storage modulus increased to 9.0 and 25.5 kPa, respectively. The yield of the product of these hydrogels was determined to be 67–99%.

The findings showed that the gelation of the hydrogels is directly related to the concentration of the crosslinking agent and the initial concentration of monomers. Gelation time, *t_gel_*, decreases as monomer or crosslinker concentration increases. The rigidity of the hydrogels approaches the maximum value of the storage modulus, G′, as the balanced charge concentration of monomers increases, which occurs because the counterions affect the mechanical and rheological characteristics of the hydrogels. At low monomer concentrations (C_M_ ≤ 15 wt%), the hydrogels are soft and fragile. At high monomer concentrations (C_M_ ≥ 15 wt%), the hydrogels show an increased G′. Raising the crosslinker content reduces gelation time and improves G′, causing the loss factor, tan δ, to decrease, which is due to the influence of electrostatic interactions between PAMPS-PAPTAC segments on the elasticity of the hydrogels.

### 3.2. Series C

PA hydrogels were synthesized using the ionic monomers PAMPS and PAPTAC at a constant molar concentration of C_M_—30 wt% in the presence of the chemical crosslinker MBAA, which was varied from 20 down to 0.1 mol.%, where decreasing cross-linker concentration improved ionic interaction, producing fewer chemical crosslinks. Hence, Series C hydrogels were synthesized.

Figure 3 shows that gelation was achieved in the hydrogels in which the crosslinking agent MBAA had been reduced to 0.25 mol.%. However, gelation was not observed in samples with a crosslinking agent content of 0.1 mol.%. As can be seen in Figure 4, the sample C30/0.1 is almost water-soluble and no longer qualifies as a true hydrogel, thus highlighting the role played by the increase in the concentration of MBAA in the improvement of the hydrogel’s mechanical properties. The best rheological properties were found in C30/5, where the *G′* is about 25.5 kPa, whereas the G′ of C30/0.25 dropped to 1.7 kPa (Figure 4). Gelation time increased (Figure 3) for hydrogels in response to a decrease in the amount of crosslinking agent, from 1 min for C30/5 to 6.5 min for C30/0.25.

An interesting point is that the hydrogels achieve high elasticity and toughness at C_M_ ≥ 15 wt%. At such a great concentration, inter-chain ionic bonds are balanced to form stable supramolecular hydrogels, even in the absence of a chemical crosslinker, thus increasing the electrostatic interaction and the mechanical performance of the polymers, as well as achieving thermodynamic equilibrium. At low C_M_, the system overcomes the electrostatic repulsion and the formation of interstrand ionic bonds is quenched, which leads to swelling of the gel. At high C_M_, inter-chain ionic compounds dominate, and the gels shrink in water. The mechanical behavior of samples in the as-prepared state and those that have undergone swelling was shown to be completely different. After prepared samples of PA hydrogels, containing a large concentration of counterions, demonstrated the typical properties of rubbery materials, including elasticity and stretchability. A greater monomer concentration in the system also results in a higher loss modulus.

Swelling kinetics studies of the hydrogels show that over approximately the first day, the gels swelled, reaching a maximum level, *m_rel_*_,*max*_. On the second day, they began a deswelling process that lasted until equilibrium was obtained, usually within 14 days (Figure 5a), creating a predominance of Coulomb repulsion in which the polymer segments elongated. As the concentrations of monomer or crosslinker were increased, the samples shrank in water, showing that the charge balance point was approaching, at which Coulomb attraction prevailed over the existing repulsion. Therefore, the polymer chains collapsed, creating a sharp increase in G′. These results show that high concentrations of monomers during polymerization are one of the main conditions for obtaining a tough gel. And at low MBAA concentrations, the system overcomes electrostatic attraction and suppresses the formation of interstrand ionic bonds, causing the gel to expand.

It was confirmed that the addition of a chemical crosslinker limited sample formation and decreased the viscoelastic properties of the gels. As shown in the present study, the conditions necessary to obtain strong PA hydrogels require the synthesis of gels at a high monomer concentration with no or only a small quantity of chemical crosslinker. A charge balance condition is required for a tough gel because the Coulomb attraction overcomes the repulsion of neutrally charged PAs.

PA hydrogels need to be synthesized at a high concentration near the charge balance point. When charge balance is achieved, they collapse, as Coulomb attraction overcomes repulsion for equally charged polyampholyte hydrogels, the IEP in this system of a PA hydrogel at the molar fraction of the monomer is achieved at the point of maximum deswelling. The entanglement of polymers is due to the substantial concentrations of monomers, resulting in superior gelation.

As a result, by decreasing the amount of cross-linker in the comonomer feeds, the hydrogels become incredibly weak. At high cross-linker concentrations, MBAA ≥ 0.5 mol.%, the hydrogels exhibited increased mechanical properties (Table 2). After the samples were immersed in water, mobile counterions were dialyzed, significantly enhancing the formation of inter-chain and intra-chain ionic bonds, which resulted in a rise in the gel fraction, *W_g_.* The creation of inter-chain hydrogen bonds, which contribute to the MBAA density of the hydrogels, could explain the increasing gel fraction [21].

The stress-strain curves of the samples where the crosslinker content was varied from 5 down to 0.1 mol.% with a fixed C_M_ = 30 wt% are presented in Figure 6a–c. Clearly, the best mechanical performance in the tensile tests was the C30/0.5 sample, where Young Modulus *E* is equal to 1.8 kPa and fracture stress σf = is equal to 7 kPa. Low MBAA concentration samples (≤0.5 mol.%) do not exhibit a yielding point, which denotes weak ionic interaction. While the gels made with a high concentration of MBAA (≥0.5 mol.%) show distinct yielding points with noticeably improved σ_f_. In both cases, the enhanced mechanical strength may be related to strong ionic bond formation at elevated crosslinker concentrations [22], while the distinct yielding indicates fragmentation due to the ionic structure (Figure 6d–f). However, the results demonstrated that the PA hydrogels had poor mechanical properties. Nonetheless, C30/0.5 was chosen because it showed good mechanical performance compared to other samples.

### 3.3. Series D: Complementary H-Bonds

In our previous study, it was shown that the mechanical properties of the material were improved by the hydrophobic interactions created by the incorporation of C18A into equimolar AMPS-APTAC hydrogels [20]. Sekizkardes et al., similarly, showed that the mechanical parameters of hydrogels containing 30% water by mass were increased by the effect of complementary H-bonds for terpolymeric gels compared to copolymeric ones [23]. In the aforementioned study, we presented a route for tunable mechanical properties by examining the electrostatic interaction and the H-bond effect together.

The hydrophilic monomer DMAA was introduced into the backbone of PA hydrogel chains to balance the effects of reduced MBAA concentration, which produced brittle hydrogels. As discussed below, even a small amount of DMAA improved gel formation. Table 2 illustrates the mechanical properties of PAs at various DMAA concentrations. To examine the effect of DMAA, the monomer concentration was fixed at C_M_ = 30 wt% in the presence of 0.5 mol.% of MBAA, whereas the amount of DMAA was varied from 5 to 20 mol.%.

Figure 7 shows the rheological characteristics of the hydrogels using the DMAA monomer. Increasing the DMAA content in the hydrogel leads to an improvement in the rheological performance of the samples. In addition, DMAA significantly decreased the swelling properties of hydrogels by up to three times compared to those without DMAA, with *W_g_* correspondingly increasing to 100% (Figure 8).

Our results show that adding DMAA to the monomer mixture drastically increases the mechanical performance of the hydrogels. Additionally, sample C30/0.5/20 shows results in Young Modulus *E* of around 22.4 kPa, which rises to 10 times as much as the same mixture without DMAA in Young Modulus *E* equal to 2 kPa. This suggests that the addition of DMAA increases the strength of the material and can therefore be used to create stronger and more durable materials. In addition, maximum elongation, ε%, did not change with an increasing amount of DMAA, which is very convenient for improving the mechanical properties of these hydrogels (Figure 9a,b).

Figure 10 shows the results of five typical tensile cycles involving the loading and unloading of specimens C30/0.5 and C30/0.5/20, where *ε_max_* was increased up to 300% for 5 cycles. The C30/0.5/20 sample showed a persistent fracture in the first cycle. However, within one minute of unloading in the second through fifth cycles, both gel samples completely and instantly returned to their previous size. The following behavior can be observed in the photographs in Figure 10c, which show the C30/0.5 and C30/0.5/20 samples being manually stretched to around 300%.

After stretching the specimens up to 300%, the samples return to their original size after 1 min of relaxation, which demonstrates that DMAA significantly contributes to the elasticity of the PAs (Figure 10a,b). With the incorporation of DMAA = 20 mol.% into the structure, the hysteresis energy *U_hys_*, was also increased.

Based on the findings, the hydrogels have a 100% self-recovery rate and are thus fatigue resistant. As seen in Figure 10b, distinct yielding, and hysteresis were detected during the first loading and unloading cycles of the C-30/0.5/20 sample. However, no change in hysteresis energy was observed for the sample without DMAA. Following a relatively short waiting time of 1 min, the stress-strain curve fully recovered to the level of the initial loading curve. It is evident from the hysteresis area change (Figure 10b), the hysteresis ratio, and the waiting time dependence of the residual strain that recovery occurs quickly, most likely due to a competition between the elasticity of the primary chain and the strength of the temporarily formed bonds during unloading [20]. The recovery is more prominent when the waiting time is short, indicating that the temporary bonds are weaker and are broken more easily. Therefore, the recovery is due to an elastic response of the primary chains and the breaking of the temporary bonds. Following elastic contraction, large deformations are observed, denoting places where bonds failed to reform after the rapid recovery. The elastic contraction is weak at minor deformations, so the reformed bonds slow the contraction of the primary chain to its equilibrium state.

Up to a loading strain of 300%, full self-recovery was observed without residual strain. This differed from the behavior of several other ionically crosslinked hydrogels, which showed permanent residual strain due to primary network damage regardless of their capacity for self-recovery, as shown in the cyclic testing, and the gels still exhibit remarkable resistance to fatigue against repeated deformation [22,24].

Table 3 and Figure 11 show the mechanical properties determined by compression tests of the samples without and with DMAA = 20 mol.% in both the after prepared (Figure 11a,c) and swollen states (Figure 11a,b). When the mechanical properties of polyampholite hydrogel samples, which are swollen and prepared, are compared, it is seen that the hydrogels synthesized in the presence of the uncharged monomer DMAA show much higher performance than the gels without DMMA. This can be explained by the regularization of the chains in DMAA-added PAs due to their hydrophilic nature or by the chain stresses resulting from higher swelling.

We also showed that the mechanical values for polyampholyte hydrogels prepared in the presence of DMAA are five times higher than those in the absence one of DMAA for both preparation (E increased from 11.7 ± 1.2 to 49.7 ± 3.9) and swollen states (E increased from 2.9 ± 0.1 to 18.1 ± 0.4) and also in the MPa scale. This increase in strength was attributed to the increased ionic interactions, hydrogen bonding, and entanglement of the polymer chains.

The current findings reveal that incorporation of the hydrophilic monomer DMAA into the backbone of PA hydrogels improves their mechanical performance, resulting in the formation of mechanically tough and stretchable hydrogels with potential applications in biological systems and industry.

## 4. Conclusions

Charged polymers have been the subject of intense investigation, both theoretically and practically, owing to their unique features and applications. They are used in a variety of applications, such as fuel cells, energy storage devices, and biomedical devices. Their properties can be tailored to a wide range of applications due to their flexibility and stability. Despite this, the mechanical properties of such hydrogels, such as their low modulus and tensile strength, prevent them from being used for load-bearing applications. For this reason, efforts have been made to improve the mechanical properties of hydrogels. Research has focused on incorporating synthetic polymers to increase the stiffness and strength of hydrogels.

In this study, through thermal polymerization in the presence of the chemical crosslinker MBAA, equimolar PAPTAC-PAMPS polyampholyte hydrogels were synthesized by free radical polymerization. These hydrogels were characterized by their rheological and mechanical properties. In order to improve the mechanical properties of the polyampholyte hydrogels, N,N-dimethylacrylamide (DMAA) was added to the structure to form a complementary H-bond. The mechanical properties of PAMPS-PAPTAC-DMAA hydrogels, especially in terms of modulus, increased approximately 21-fold compared to gels prepared without DMAA. The results show that the crucial factors determining the mechanical performance of the copolymer hydrogels are the H-bonding cooperativity via incorporation of DMAA segments into the physical gels and hence electrostatic interactions.

The application of hydrogels was not the focus of our work; however, we believe that equimolar polyampholytes containing both positively and negatively charged functional groups demonstrate remarkable toughness and stretchability due to hydrogen bonding resulting from the addition of DMAA to the system. These hydrogels can act as artificial muscles, providing actuation and movement in soft robotic systems. Their stretchability and toughness enable robots to perform complex movements and withstand external forces.

## Data Availability

Not applicable.

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
