# Peer review of "Equimolar Polyampholyte Hydrogel Synthesis Strategies with Adaptable Properties"

_polymers, 2023, doi:10.3390/polym15143131_

Round 1

Reviewer 1 Report

This work aims to investigate the effect of amount of monomer and crosslinker on the mechanical and rheological properties of equimolar polyampholyte hydrogels. The topic is both relevant and interesting for the research community, as the polyampholyte hydrogels have interesting potential biomedical applications. In my opinion, the manuscript could be accepted for publication after minor modifications:

a) In page 2, line 62 the authors use the acronym DMAA which has not been previously defined.

b) In page 2, line 76 the authors state that the chemical crosslinker was varied from 20 to 0.1%mol. These values does not match the information presented in Table 1.

c)       In page 2, line 65 the authors start the paragraph with a number: “4 sets of PA hydrogels”.

d)      In page 5, line 155 the authors state the the gelation time for samples C%/20 and C%/10 was 9.6 and 31.1 min, respectively. It is impossible to see these values clearly in Figure 1.

e)      In page 5, the figure 1e includes an inset plot, but it lacks the scale in both the vertical and horizontal axis.

f)        In page 7, the caption of figure 3 does not describe what is the concentration of crosslinker corresponding to each figure from figure 3a to figure 3e.

g)       In page 8, the caption of figure 4a to figure 4e does not describe the difference between each sub-figure.

h)   Page 14. References 5, and 26 should be reviewed as they are missing either the page number of the cite or the title of the work.

The quality of english is good.

Author Response

Dear Referee,
The answers prepared for your questions are attached.

Kind regards..

Reviewer 2 Report

In the present manuscript Toleutay et al. describe the synthesis of different series of equimolar polyampholyte hydrogels with variable anionic (acrylamidomethylpropane sulfonic acid sodium salt) and cationic (acrylamidopropyl-trimethylammonium chloride) monomers content. The authors assess the mechanical and rheological properties of the obtained hydrogels in correlation with changes in monomer concentration and crosslinker ratios. Despite the paper is displaying some interesting results, it should be revised according to the following comments:

Comment 1:

The authors provided schematic illustration of the polyampholites. It could be useful also to provide the chemical structures of the different components of the hydrogel.

Can the prepared gel’s network be visualized using microscopy analysis?

Comment 2:

Although the study contains a considerable number of tests and results, the discussion of these results according to the literature is very poor. In the discussion section there is no comparison of the obtained results with those of other works.

Comment 3:

The authors should clarify the potential application(s) of the hydrogels they developed; they only mentioned some general applications. How the mechanical properties of these materials can influence their future applications/use. 

Comment 4:

Although conclusion section is optional, I suppose it should be added here to highlight the main findings of the study.

Author Response

(The authors gave the same response as above.)

Round 2

Reviewer 2 Report

I believe that the authors have improved the manuscript. The latter could be accepted for publication after considering the following:

- The discussion of the obtained results in comparison with other works is still a weak side of the study and can be further improved. If there are few works related to this type of hydrogels, this should be mentioned.

- The newly added conclusion section is too lengthy. Only the main findings should be highlighted in "conclusions". The remaining details can be combined with (returned to) the "results and discussion" section.

Author Response

Dear referee,

Thank you for your encouragement and hard work. The article has been re-edited in line with your suggestions given below.

I believe that the authors have improved the manuscript. The latter could be accepted for publication after considering the following:

- The discussion of the obtained results in comparison with other works is still a weak side of the study and can be further improved. If there are few works related to this type of hydrogels, this should be mentioned.

Answer: With the recommendation of the referee, the literature summary was expanded by giving the details of the study in the literature. A recently published review is also included (Please see reference 12).

- The newly added conclusion section is too lengthy. Only the main findings should be highlighted in "conclusions". The remaining details can be combined with (returned to) the "results and discussion" section.

Answer: With the suggestion of the referee, Conclusion was rearranged to include only the starting point of the study and the conclusions reached.